# Serological evidence of West Nile virus circulation in Indonesia: Initial insights into an underrecognized flavivirus threat

Yora Permata Dewi[1], Rintis Noviyanti[2], Leily Trianty[2], Enny Kenangalem[3,4], Frilasita Aisyah Yudhaputri[1], Matthew J. Grigg[5], Jeremy P. Ledermann[6], Ann M. Powers[6], Eric C. Mossel[6], Khin Saw Aye Myint[1]*

1 Exeins Health Initiative, Jakarta, Indonesia, 2 Eijkman Research Center for Molecular Biology, The National Research and Innovation Agency, Cibinong, Indonesia, 3 Timika Malaria Research Program, Papuan Health and Community Development Foundation, Timika, Indonesia, 4 Rumah Sakit Umum Daerah Kabupaten Mimika, Timika, Indonesia, 5 Global and Tropical Health Division, Menzies School of Health Research and Charles Darwin University, Darwin, Australia, 6 Centers for Disease Control and Prevention, Fort Collins, Colorado, United States of America

* khinsawying@hotmail.com

## Abstract

West Nile virus (WNV) is a mosquito-borne flavivirus primarily transmitted by *Culex* species and maintained in a bird–mosquito–bird cycle. Though WNV is not considered endemic to Indonesia, sporadic molecular and serological findings suggest possible under-recognized circulation. To examine this possibility, we conducted a cross-sectional serosurvey of 569 archived serum and plasma samples collected between 2021 and 2023 from non-febrile individuals, including blood donors in Papua and community-based participants in Kalimantan. Samples were initially screened using a commercial WNV IgM Enzyme Linked Immunosorbent Assay (ELISA), IgM-positive and equivocal samples were further tested with IgM ELISAs against closely related flaviviruses, including dengue virus (DENV), Japanese encephalitis virus (JEV), and Zika virus (ZIKV). WNV-reactive IgM was initially detected in 9 (1.58%) samples, with 7 (1.23%) additional equivocal results. After additional testing, the WNV IgM seroprevalence was 0.35%, while 1.76% showed cross-reactivity to other flaviviruses. These cases spanned regions including North Kalimantan and Papua, where evidence of WNV-related exposure was previously reported. Detection of WNV-reactive IgM in asymptomatic individuals suggests potential silent or undetected circulation of WNV in Indonesia. Given diagnostic cross-reactivity and IgM persistence, further studies using molecular tools, plaque reduction neutralization test (PRNT), IgG testing, and entomological data are needed to clarify WNV epidemiology and inform surveillance strategies in the region.

**Data availability statement:** All relevant data are within the paper and its Supporting Information files.

**Funding:** This work was supported by the U.S. Centers for Disease Control and Prevention [award numbers U18CK000443 and U01CK000577] granted to KSM, and funded through the Australian Centre for International Agricultural Research and the Indo-Pacific Centre for Health Security, Department of Foreign Affairs and Trade, Australian Government [grant number LS/2019/116] awarded to MJG. Additional support was provided by the Exeins Health Initiative through institutional research funds. The funders had no role in study design, data collection and analysis, decision to publish, or preparation of the manuscript.

**Competing interests:** The author(s) have declared no potential conflicts of interest with respect to the research, authorship, and/or publication of this article. The findings and conclusions in this report are those of the authors and do not necessarily represent the views of the Centers for Disease Control and Prevention.

## Introduction

West Nile virus (WNV) is a mosquito-borne flavivirus belonging to the Japanese encephalitis virus (JEV) serocomplex [1]. It is primarily transmitted by *Culex* mosquitoes and maintained in an enzootic cycle involving avian hosts as amplifying reservoirs. First identified in Uganda in 1937 [2], WNV has since emerged as a significant global health concern. Since the 1990s, it has progressively spread across all continents, with notable outbreaks in the United States marked by severe neurological disease [3]. Most human infections (approximately 80%) are asymptomatic, while a minority present with clinical manifestations ranging from mild febrile illness to severe neuro-invasive disease, including meningitis, encephalitis, and acute flaccid myelitis [4]. The high frequency of inapparent infections in humans complicates surveillance efforts and contributes to substantial underreporting.

WNV is not considered endemic in Indonesia. However, evidence of past exposure of a sub-type of WNV, Kunjin Virus (KUNV), has been documented through serological studies in indigenous populations from Kalimantan, including Pontianak, Samarinda, and Balikpapan, as well as in Jayapura (Papua) [5]. Additional seropositivity has been reported in both humans and animals in Lombok [6]. In 2014, WNV RNA was detected in a febrile patient from Bandung [7], providing molecular evidence suggestive of recent viral circulation. Yet, such findings remain sparse and geographically limited. The apparent near absence of confirmed human cases may reflect diagnostic limitations and surveillance gaps rather than a true lack of viral activity. Moreover, the co-circulation of other flaviviruses, such as dengue virus (DENV) and JEV, poses significant diagnostic challenges, increasing the likelihood that WNV infections are under-recognized or misclassified.

This study aimed to investigate the presence of WNV infections among non-febrile individuals. Relying solely on surveillance of febrile or hospitalized patients risks overlooking the broader scope of WNV transmission. Understanding the true extent of infection in the community, including regions such as Papua and North Kalimantan, is essential for accurate risk assessment and guiding public health interventions.

## Methods

Ethical approval for this study was obtained from the Universitas Katolik Indonesia Atma Jaya Ethics Committee (No: 001F/III/PPPE.PM.10.05/02/2025) and Research Ethics Committee on Health, Hasanuddin University Hospital (No. 840/UN4.6.4.5.31/PP36/2024). This study analyzed archived, anonymized serum and plasma samples that were originally collected between 2021 and 2023 across five regions in Indonesia: Jayapura (Papua), Timika (Central Papua), Manokwari (West Papua), and Malinau and Nunukan (North Kalimantan) (Fig 1).

In Papua, samples were obtained from adult blood donors through the Indonesian Red Cross blood bank. In North Kalimantan, samples were collected from community-based participants, including both adults and children. Written informed consent had been obtained during the original sample collection from all adult participants and from the legal guardians of minors.

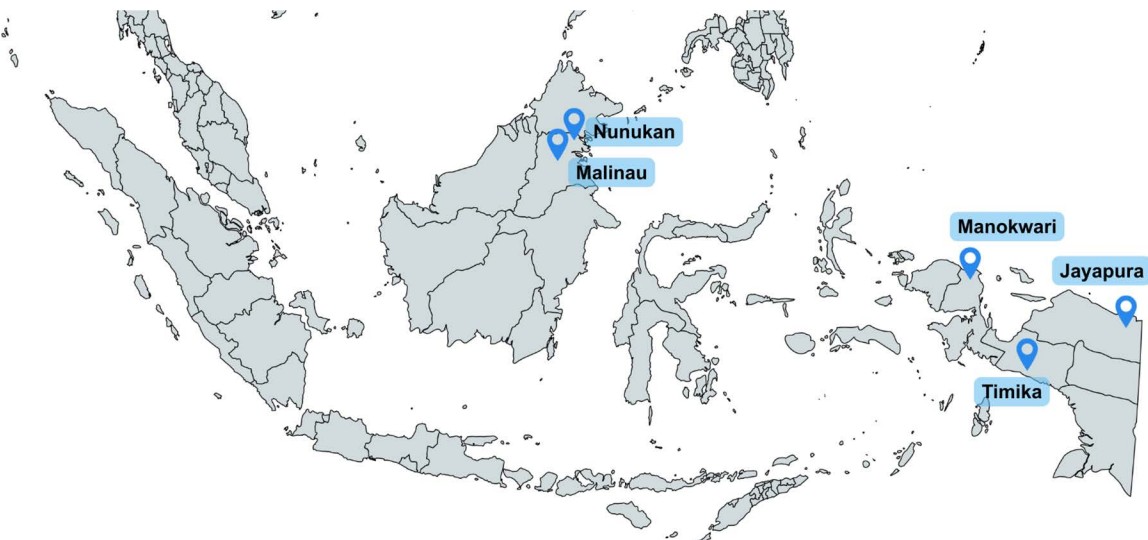

**Fig 1. Study sites for the West Nile virus serosurvey in Indonesia: Malinau and Nunukan (North Kalimantan), Manokwari (West Papua), Timika (Central Papua), and Jayapura (Papua).** Map generated using MapChart (https://mapchart.net). Republished under a CC BY 4.0 license, with permission from Minas Giannekas (Founder and copyright holder of MapChart), original copyright © MapChart 2025.

For the current analysis, all archived samples were fully anonymized prior to access, and researchers had no access to identifiable information at any stage. The Ethics Committee waived the requirement for additional informed consent for this secondary use of de-identified materials. Archived samples and associated data were accessed for research purposes in May 2025.

Samples were screened for WNV using a commercial qualitative anti-IgM Enzyme-linked Immunosorbent Assay (ELISA) kits (InBios International Inc., Seattle, WA), with a reported sensitivity of 96.2–99.4% and specificity 98.9–100% [8]. For this assay, the optical density (OD) of each sample against recombinant WNV antigen was divided by the OD of the negative-control antigen to generate an immune status ratio (ISR). ISR values were interpreted as follows: <4.47 = negative, 4.47–5.66 = equivocal, and >5.66 = positive. Equivocal (EQ) results were retested using the same WNV kit. Samples that tested negative for WNV IgM were not tested further.

All samples that were found to be WNV IgM-positive or EQ were subsequently tested using IgM ELISA for DENV, JEV, and Zika virus (ZIKV), using CDC MAC-ELISA protocols [9]. For this assay, the mean OD of each test serum (P) was divided by the mean OD of the corresponding negative-control serum (N) to generate a P/N ratio. Results were interpreted as positive (P/N ≥3.0), equivocal (P/N ≥2.0 and <3.0), or negative (P/N <2.0). Final serostatus was interpreted based on the combination of assay results and was determined as follows:

• WNV positive: positive WNV IgM with negative results for all other flavivirus IgM tests.

• Flavivirus-positive: WNV IgM-positive with a positive or EQ result for ≥1 additional flavivirus (DENV, JEV, or ZIKV).

• Negative: WNV IgM-negative samples, which were not tested further.

All analyses were descriptive. Participant characteristics, including age group, sex, and study site, were summarized using frequencies and percentages. Age was analyzed only as a categorical variable. The overall WNV IgM seroprevalence was calculated as the proportion of confirmed WNV IgM–positive samples among all tested samples. Site-specific seroprevalence estimates were calculated descriptively; no statistical comparisons between sites were performed, due to

the small number of confirmed WNV-positive samples. Secondary outcomes included the proportions of samples classified as flavivirus-positive and equivocal for WNV IgM. No hypothesis testing or inferential statistical analyses were conducted. All statistical summaries were performed using Microsoft Excel. The anonymized dataset underlying this study, including sex, age group, and final serostatus for all samples analyzed, is provided as Supporting Information (S1 Data).

## Results

The mean participant age was 33.4 years. There were more males (*n* = 323, 56.8%) compared with females (*n* = 246, 43.2%). Most individuals were in the 18–32 age bracket (50.4%) (Table 1).

A total of 569 samples from four provinces were tested for evidence of WNV infection. Of these, 9 (1.58%) had detectable IgM antibody and 7 (1.23%) remained equivocal for WNV IgM even after retesting. All 16 samples were subsequently tested using IgM ELISAs against closely related flaviviruses. Following this testing, only 2 out of 569 samples were confirmed positive for WNV IgM, yielding an estimated seroprevalence of 0.35%. These positive cases were from Central Papua and North Kalimantan. Ten samples were interpreted as flavivirus-positive due to cross-reactivity with other flaviviruses. Four samples were interpreted as equivocal WNV IgM with no detectable reactivity to the other flaviviruses tested. Interpretation of individual results is shown in Table 2.

## Discussion

Although WNV has been previously reported in Indonesia through molecular and serological findings [5–7], available data remain limited, geographically fragmented, and outdated. This study contributes updated serologic evidence of possible WNV exposure among non-febrile individuals, a group often underrepresented in arbovirus surveillance. The detection of WNV-specific IgM antibodies in asymptomatic participants suggests the possibility of under-recognized viral circulation. Because most WNV infections are mild or asymptomatic, they are easily missed by passive surveillance, potentially obscuring the true burden of disease.

Our findings expand the current geographic evidence by providing serological indications of WNV exposure in both Kalimantan and Papua. Earlier studies detected low-frequency seroreactivity to KUNV, a subtype of WNV, though the interpretation was constrained by the use of hemagglutination inhibition assays and potential antigenic cross-reactivity or sequential exposure to other group B arboviruses [5]. Our findings provide updated evidence suggesting the possibility of ongoing, yet overlooked, WNV circulation in these areas.

**Table 1. Demographic characteristics of participants by study site.**

| Parameter | | North Kalimantan | | Papua | | |
|---|---|---|---|---|---|---|
| | Overall, n (%) | Nunukan | Malinau | Papua | Central Papua | West Papua |
| *N* | 569 | 125 | 124 | 70 | 125 | 125 |
| | | | | | | |
| Sex | | | | | | |
| Male | 323 (56.8) | 38 (30) | 55 (44) | 54 (77) | 85 (68) | 91 (73) |
| Female | 246 (43.2) | 87 (70) | 69 (56) | 16 (23) | 40 (32) | 34 (37) |
| | | | | | | |
| Age group | | | | | | |
| 3-17 | 18 (3.2) | 5 (4.0) | 12 (9.7) | 1 (1.4) | 0 (0.0) | 0 (0.0) |
| 18-32 | 286 (50.4) | 66 (52.8) | 54 (43.5) | 39 (55.7) | 49 (39.2) | 78 (62.9) |
| 33-47 | 187 (32.9) | 31 (24.8) | 30 (24.2) | 26 (37.1) | 59 (47.2) | 41 (33.1) |
| 48-62 | 67 (11.8) | 21 (16.8) | 20 (16.1) | 4 (5.7) | 17 (13.6) | 5 (4.0) |
| 63-77 | 10 (1.8) | 2 (1.6) | 8 (6.5) | 0 (0.0) | 0 (0.0) | 0 (0.0) |

**Table 2. Final interpretation of WNV seropositivity based on screening WNV IgM ELISA followed by differential IgM ELISAs to rule out cross-reactivity for other co-circulating flaviviruses.**

| No | City | Province | Sex | Age | WNV IgM | | DENV IgM | | JEV IgM | | ZIKV IgM | | Final interpretation |
|----|------|----------|-----|-----|---------|---------|----------|---------|---------|---------|----------|---------|----------------------|
| | | | | | ISR | Interpretation | P/N | Interpretation | P/N | Interpretation | P/N | Interpretation | |
| 1 | Timika | Central Papua | Male | 37 | 6.20 | + | 1.17 | – | 1.07 | – | 1.20 | – | WNV-positive |
| 2 | Manokwari | West Papua | Male | 34 | 6.89 | + | 0.99 | – | 4.48 | + | 1.41 | – | Flavivirus-positive |
| 3 | Malinau | North Kalimantan | Male | 73 | 8.71 | + | 1.19 | – | 1.79 | – | 2.53 | EQ | Flavivirus-positive |
| 4 | Malinau | North Kalimantan | Female | 48 | 58.02 | + | 16.21 | + | 28.40 | + | 16.30 | + | Flavivirus-positive |
| 5 | Malinau | North Kalimantan | Male | 29 | 9.19 | + | 10.83 | + | 4.38 | + | 22.24 | + | Flavivirus-positive |
| 6 | Nunukan | North Kalimantan | Female | 40 | 13.53 | + | 1.12 | – | 6.65 | + | 3.21 | + | Flavivirus-positive |
| 7 | Nunukan | North Kalimantan | Female | 20 | 8.38 | + | 1.83 | – | 3.24 | + | 2.68 | EQ | Flavivirus-positive |
| 8 | Nunukan | North Kalimantan | Female | 20 | 6.48 | + | 1.58 | – | 2.47 | EQ | 2.69 | EQ | Flavivirus-positive |
| 9 | Nunukan | North Kalimantan | Male | 34 | 12.16 | + | 1.40 | – | 1.84 | – | 1.27 | – | WNV-positive |
| 10 | Jayapura | Papua | Male | 39 | 5.03 | EQ | 3.57 | + | 2.25 | EQ | 1.76 | – | Flavivirus-positive |
| 11 | Manokwari | West Papua | Male | 23 | 4.94 | EQ | 1.05 | – | 1.10 | – | 1.10 | – | WNV-equivocal |
| 12 | Manokwari | West Papua | Male | 27 | 5.26 | EQ | 2.19 | EQ | 2.38 | EQ | 1.61 | – | Flavivirus-positive |
| 13 | Malinau | North Kalimantan | Female | 18 | 5.21 | EQ | 18.98 | + | 16.04 | + | 10.38 | + | Flavivirus-positive |
| 14 | Nunukan | North Kalimantan | Male | 13 | 4.61 | EQ | 1.11 | – | 0.46 | – | 1.26 | – | WNV-equivocal |
| 15 | Nunukan | North Kalimantan | Female | 33 | 5.31 | EQ | 1.10 | – | 1.09 | – | 1.18 | – | WNV-equivocal |
| 16 | Nunukan | North Kalimantan | Male | 56 | 5.44 | EQ | 0.99 | – | 0.80 | – | 1.42 | – | WNV-equivocal |

Note: Interpretation for WNV IgM: Immune status ratio (ISR) <4.47: negative, ISR ≥4.47 to <5.66: equivocal, ISR >5.66: positive; Interpretation for CDC MAC-ELISA: P/N <2: negative, P/N ≥2 to <3: equivocal, P/N ≥3: positive; +: positive, EQ: equivocal, -: negative.

These findings also align with regional evidence of WNV circulation in Southeast Asia. In Peninsular Malaysia, WNV-reactive antibodies have been reported in wild birds, livestock, horses, and humans [10–12], suggesting ongoing zoonotic exposure. In Sarawak, Malaysian Borneo, a WNV-like virus was isolated from *Culex pseudovishnui* mosquitoes as early as 1970 [11]. Given the proximity of North Kalimantan to Sarawak and the ecological continuity of *Culex* vector and bird migration routes, our detection of WNV IgM in humans further supports the likelihood of low-level or previously undetected transmission in Indonesian Borneo.

In our study, several samples were initially positive for WNV IgM; however, only two remained positive after additional testing for closely related viruses, suggesting true WNV-specific responses. The use of multiple IgM assays effectively increased the specificity of detection and strengthens the interpretation that these individuals were exposed to WNV rather than exhibiting cross-reactive responses.

Although the commercial ELISA used in this study is reported to have high sensitivity and specificity, interpretation in flavivirus-endemic settings remains challenging. Indonesia, where DENV, JEV, and ZIKV co-circulate extensively, represents such a setting, as strong antigenic similarities among flaviviruses often lead to cross-reactivity [13]. This was evident in a subset of samples that tested positive for multiple flaviviruses, which may reflect either multiple flavivirus infections or cross-reactive responses from prior exposures. In such context, differential ELISAs could serve as a valuable confirmatory tool, offering higher specificity to distinguish true WNV infection from cross-reactive responses to other flaviviruses.

Additionally, we observed equivocal WNV IgM reactivity in samples that tested negative for DENV, JEV, and ZIKV suggesting these borderline signals were not the result of serologic cross-reactivity with the major co-circulating flaviviruses tested. Such findings may reflect low-level or waning immune responses from prior WNV exposure, exposure to a flavivirus not tested for, or alternatively, nonspecific assay reactivity [14]. The absence of concurrent reactivity to other

known regional flaviviruses strengthens the specificity of the WNV signal; however, the lack of symptoms and confirmatory testing limits definitive interpretation. Equivocal ELISA results are not uncommon and should be interpreted cautiously, particularly in populations with high flavivirus exposure risk but no clinical evidence of acute infection.

Furthermore, the detection of WNV IgM in non-febrile individuals raises the issue of IgM persistence. Although IgM is commonly used as a marker of recent infection in surveillance studies, as in other reports, evidence exists demonstrating that WNV-specific IgM antibodies can persist for months or even years post-infection. Because our study was limited to single serum samples, we could not distinguish whether IgM positivity indicated recent infection or past exposure to WNV. Determining this would require additional diagnostic methods such as paired acute and convalescent sera to assess IgG seroconversion, which was not feasible with the available samples.

This study has several limitations. The absence of PRNT confirmation prevents us from fully excluding the possibility of flavivirus cross-reactivity. IgG testing was not included due to resource and logistical constraints, limiting our ability to differentiate recent from past exposure or to assess cumulative seroprevalence. The very small number of positive cases limits the interpretability of our findings and precludes detection of any regional patterns. Furthermore, the cross-sectional design captures data at a single time point, limiting inferences about transmission dynamics.

## Conclusion

This study provides updated serological evidence of WNV exposure among non-febrile individuals from Kalimantan and Papua, extending the geographic footprint of WNV in Indonesia. Although confirmatory testing was not available, the findings support the hypothesis of silent or under-recognized WNV circulation across multiple regions of Indonesia. Given the diagnostic complexities inherent in flavivirus serology, future research should incorporate molecular testing, PRNT confirmation, IgG testing, and integrated ecological surveillance to better understand WNV epidemiology and guide public health strategies in Indonesia.

## Supporting information

**S1 Data. Anonymized dataset containing sex, age group, and final serostatus.**
(XLSX)

## Acknowledgments

We acknowledge dr. Raflus Doranggi and dr. Christina for their exceptional institutional support and coordination, which allowed effective community access and sample collection. We also sincerely thank the Emerging Virus Team at Exeins Health Initiative for their dedicated efforts and continuous coordination throughout the study. We thank the participants and their families for agreeing to participate, as well as the staff of Indonesian Red Cross Society (PMI) Papua, Papua Barat, and Mimika, Mitra Masyarakat Hospital Timika, and Timika Distric Hospital (RSUD Timika) for their support.

## Author contributions

**Conceptualization:** Rintis Noviyanti, Ann M. Powers, Eric C. Mossel.

**Data curation:** Yora Permata Dewi, Leily Trianty, Enny Kenangalem.

**Formal analysis:** Yora Permata Dewi, Eric C. Mossel.

**Funding acquisition:** Rintis Noviyanti, Matthew J. Grigg, Khin Saw Aye Myint.

**Investigation:** Yora Permata Dewi.

**Methodology:** Yora Permata Dewi, Eric C. Mossel, Khin Saw Aye Myint.

**Project administration:** Leily Trianty, Frilasita Aisyah Yudhaputri.

**Resources:** Rintis Noviyanti, Leily Trianty, Enny Kenangalem, Frilasita Aisyah Yudhaputri, Matthew J. Grigg, Jeremy P. Ledermann, Ann M. Powers, Eric C. Mossel, Khin Saw Aye Myint.

**Supervision:** Rintis Noviyanti, Leily Trianty, Enny Kenangalem, Matthew J. Grigg, Ann M. Powers, Eric C. Mossel, Khin Saw Aye Myint.

**Validation:** Jeremy P. Ledermann, Ann M. Powers, Eric C. Mossel, Khin Saw Aye Myint.

**Visualization:** Yora Permata Dewi.

**Writing – original draft:** Yora Permata Dewi.

**Writing – review & editing:** Yora Permata Dewi, Rintis Noviyanti, Leily Trianty, Enny Kenangalem, Frilasita Aisyah Yudhaputri, Matthew J. Grigg, Jeremy P. Ledermann, Ann M. Powers, Eric C. Mossel, Khin Saw Aye Myint.

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
