## [Decision Letter · Decision Letter 0]

19 Nov 2025

Dear Dr. Myint,

Thank you for submitting your manuscript to PLOS ONE. After careful consideration, we feel that it has merit but does not fully meet PLOS ONE’s publication criteria as it currently stands. Therefore, we invite you to submit a revised version of the manuscript that addresses the points raised during the review process.

In particular, the reviewers were positive about the overall merits of the manuscript, but they also pointed out several aspects, especially within the methodology section, requiring further clarification. Please address the reviewers’ comments carefully and submit your revised manuscript by Jan 03 2026 11:59PM. If you will need more time than this to complete your revisions, please reply to this message or contact the journal office at plosone@plos.org . A rebuttal letter that responds to each point raised by the academic editor and reviewer(s). You should upload this letter as a separate file labeled 'Response to Reviewers'.A marked-up copy of your manuscript that highlights changes made to the original version. You should upload this as a separate file labeled 'Revised Manuscript with Track Changes'.An unmarked version of your revised paper without tracked changes. You should upload this as a separate file labeled 'Manuscript'.

We look forward to receiving your revised manuscript.

Kind regards,

Mirko Cortese

Academic Editor

PLOS ONE

“This work was supported by the U.S. Centers for Disease Control and Prevention [award numbers U18CK000443 and U01CK000577] granted to KSM, and funded through the Australian Centre for International Agricultural Research and the Indo-Pacific Centre for Health Security, Department of Foreign Affairs and Trade, Australian Government [grant number LS/2019/116] awarded to MJG. Additional support was provided by the Exeins Health Initiative through institutional research funds. The funders had no role in study design, data collection and analysis, decision to publish, or preparation of the manuscript.”

“This work was supported by the U.S. Centers for Disease Control and Prevention [award numbers U18CK000443 and U01CK000577] granted to KSM, and funded through the Australian Centre for International Agricultural Research and the Indo-Pacific Centre for Health Security, Department of Foreign Affairs and Trade, Australian Government [grant number LS/2019/116] awarded to MJG. Additional support was provided by the Exeins Health Initiative through institutional research funds. The funders had no role in study design, data collection and analysis, decision to publish, or preparation of the manuscript.”

Reviewers' comments:

Reviewer's Responses to Questions

**Comments to the Author**

1. Is the manuscript technically sound, and do the data support the conclusions?

Reviewer #1: Partly

Reviewer #2: Yes

2. Has the statistical analysis been performed appropriately and rigorously?

Reviewer #1: N/A

Reviewer #2: No

3. Have the authors made all data underlying the findings in their manuscript fully available?

Reviewer #1: Yes

Reviewer #2: Yes

4. Is the manuscript presented in an intelligible fashion and written in standard English?

Reviewer #1: Yes

Reviewer #2: Yes

Reviewer #1: This paper provided evidence of West Nile Virus IgM positivity among people in Kalimantan and Papua. Unfortunately, there are some points that should be addressed to improve the paper quality.

Comments

Methods:

Fourth paragraph, line 102 – 104: Add the reference for the reported sensitivity and specificity for the test.

Fourth paragraph, line 108 – 109: Add more information on how to decide the final serostatus.

Results:

Table 1: The numbers for N in Nunukan and Malinau sites were not matched with the numbers for sex and age group. Check again the numbers.

Table 2: The notes under the table should be added in the Methods section and in more detail.

Discussion:

Line 173 – 183 : Are there any references for this paragraph?

Last paragraph, line 191 – 193: “The modest sample size limits statistical power…” This study did not have “modest sample size”, it had “very small number of positive cases”. The authors did not mention anything about wanted to prove something by statistical test. Hence, it is quite confusing why suddenly the statistical power was mentioned as study limitation.

With two positive cases, it was no longer “reduces the ability to detect regional patterns”. The authors did not have the ability to detect regional patterns.

Reviewer #2: This is a well written and valuable study that provides updated serologic data on WNV exposure in a population often underrepresented in surveillance. The findings are timely and contribute to understanding possible WNV circulation in Indonesia. For future, inclusion of IgG testing and addition of negative serum tested for JEV sero-complexes could strengthen the assessment and make the study’s design and analysis more robust and informative.

Introduction

Line 60 – to be more precise and use commonly used term in current literature, acute flaccid myelitis

Line 61- to be more specific, the surveillance in humans only or including mosquitoes

Line 74- It was stated to investigate recent infection, but some of the samples were old sample from 3 years ago, the word recent infection is not suitable. It is understandable your focus in on non-febrile patient, therefore this sentence needs to be rephrased.

Line 75 – specify the general population location

Line 76- state the study focuses on non-febrile patient.

Methodology

Line 104 - state what type of ELISA. How the cut-off value of ISR is determined for both ELISA (WNV IgM and CDC MAC -ELISA)? The result is an OD or concentration of antibody?

Line 108 - where did you perform the CDC MAC-ELISA protocol?

State possible reason why IgG based ELISA and RT-PCR was not included in this study?

Add statistical analysis.

Discussion

While previous studies shown that WNV IgM can persist and sometimes indicate exposure, its duration is variable and less predictable in mild or asymptomatic infections. For more robust evidence of prior or cumulative exposure, including an IgG-based ELISA is recommended, ideally alongside IgM to distinguish recent versus past infection. Also how to confirm that the infection is recent if IgM-based ELISA could indicate both recent and exposure based on the study objective ( Line 74).

Have you considered screening serum samples from febrile patients that tested negative for dengue, Zika and JE? This could be a valuable approach for future studies.

**Do you want your identity to be public for this peer review?** For information about this choice, including consent withdrawal, please see our Privacy Policy

Reviewer #1: No

Reviewer #2: No

---

## [Author Response · Author response to Decision Letter 1]

9 Jan 2026

Reviewer #1: This paper provided evidence of West Nile Virus IgM positivity among people in Kalimantan and Papua. Unfortunately, there are some points that should be addressed to improve the paper quality.

RESPONSE: We appreciate the reviewer’s constructive comments highlighting important issues and have revised accordingly, as detailed below.

Comments

Methods:

Fourth paragraph, line 102 – 104: Add the reference for the reported sensitivity and specificity for the test.

RESPONSE: A reference was added to reflect the revision.

Fourth paragraph, line 108 – 109: Add more information on how to decide the final serostatus.

RESPONSE: We appreciate the reviewer’s suggestion and have added a detailed explanation of how final serostatus was determined. These definitions are now included in the Methods section (Lines 121-126).

Results:

Table 1: The numbers for N in Nunukan and Malinau sites were not matched with the numbers for sex and age group. Check again the numbers.

RESPONSE: We thank the reviewer for spotting this typographical error. The values have been corrected.

Table 2: The notes under the table should be added in the Methods section and in more detail.

RESPONSE: It has now been integrated into the Methods section for clarity.

Discussion:

Line 173 – 183: Are there any references for this paragraph?

RESPONSE: Relevant reference has now been added to support this section.

Last paragraph, line 191 – 193: “The modest sample size limits statistical power…” This study did not have “modest sample size”, it had “very small number of positive cases”. The authors did not mention anything about wanted to prove something by statistical test. Hence, it is quite confusing why suddenly the statistical power was mentioned as study limitation.

With two positive cases, it was no longer “reduces the ability to detect regional patterns”. The authors did not have the ability to detect regional patterns.

RESPONSE: We thank the reviewer for this comment and agree that the original wording overstated our ability to perform statistical analyses. We have made the necessary correction (Lines 233-234).

Reviewer #2: This is a well written and valuable study that provides updated serologic data on WNV exposure in a population often underrepresented in surveillance. The findings are timely and contribute to understanding possible WNV circulation in Indonesia. For future, inclusion of IgG testing and addition of negative serum tested for JEV sero-complexes could strengthen the assessment and make the study’s design and analysis more robust and informative.

RESPONSE: We thank Reviewer 2 for the positive comments.

Introduction

Line 60 – to be more precise and use commonly used term in current literature, acute flaccid myelitis

RESPONSE: The term has been revised to acute flaccid myelitis.

Line 61- to be more specific, the surveillance in humans only or including mosquitoes

RESPONSE: The sentence has been revised to “The high frequency of inapparent infections in humans complicates surveillance efforts and contributes to substantial underreporting.”

Line 74- It was stated to investigate recent infection, but some of the samples were old sample from 3 years ago, the word recent infection is not suitable. It is understandable your focus in on non-febrile patient, therefore this sentence needs to be rephrased.

RESPONSE: We have revised the statement to “This study aimed to investigate the presence of WNV infections among non-febrile individuals.”

Line 75 – specify the general population location

RESPONSE: This has been added (Line 78). The detailed location of the general population sampled is described in the Methods section (Paragraph 1).

Line 76- state the study focuses on non-febrile patient.

RESPONSE: This has been clarified in the revised text.

Methodology

Line 104 - state what type of ELISA. How the cut-off value of ISR is determined for both ELISA (WNV IgM and CDC MAC -ELISA)? The result is an OD or concentration of antibody?

RESPONSE: We have revised the Methods section to specify the type of ELISA used and to explain how the cut-off values for both the InBios WNV IgM ELISA and the CDC MAC-ELISA were determined (Lines 107-129).

Line 108 - where did you perform the CDC MAC-ELISA protocol?

State possible reason why IgG based ELISA and RT-PCR was not included in this study?

Add statistical analysis.

RESPONSE: All laboratory assays, including the CDC MAC-ELISA, were performed at our laboratory at the Exeins Health Initiative (EHI), Indonesia.

IgG ELISA and RT-PCR were not included in this study due to the study’s focus on detecting IgM responses in non-febrile individuals rather than assessing long-term immunity (IgG) or active viremia (RT-PCR), combined with the limited diagnostic yield expected from the available specimens and budgetary constraints that precluded additional testing.

Statistical analysis has now been added in Lines 131-140.

Discussion

While previous studies shown that WNV IgM can persist and sometimes indicate exposure, its duration is variable and less predictable in mild or asymptomatic infections. For more robust evidence of prior or cumulative exposure, including an IgG-based ELISA is recommended, ideally alongside IgM to distinguish recent versus past infection. Also how to confirm that the infection is recent if IgM-based ELISA could indicate both recent and exposure based on the study objective (Line 75-76).

RESPONSE: We thank the reviewer for this important clarification. In the revised Discussion, we clarify that determining truly recent WNV infection was not possible in this study because only single serum samples were available (Lines 224-228).

We also agree that IgG-based ELISA, ideally interpreted together with IgM, would provide more robust evidence of cumulative or past exposure and help distinguish recent from historic infection. However, IgG testing was not included in the present study due to resource and logistical constraints, and we now explicitly acknowledge this as a limitation (Lines 230-233). We have revised the study objective and related text (Line 75) to avoid implying that IgM alone can confirm recent infection. The updated wording now states that the study aimed to investigate the presence of WNV infections among non-febrile individuals (Lines 75-76), rather than strictly “recent” infections.

Confirming recent infection would require additional diagnostic approaches such as demonstrating IgG seroconversion using paired acute and convalescent sera. These methods were beyond the scope of the current study and were not feasible given the available samples. This clarification ensures that the interpretation of IgM results remains accurate and does not overstate what IgM-based ELISA can determine.

Have you considered screening serum samples from febrile patients that tested negative for dengue, Zika and JE? This could be a valuable approach for future studies.

RESPONSE: This approach was not feasible for the present study, which focused exclusively on non-febrile individuals, and access to febrile samples fell outside the project scope. We have added this as a recommendation for future studies.

---

## [Editor Report · Decision Letter 1]

26 Jan 2026

Serological evidence of West Nile virus circulation in Indonesia: Initial insights into an underrecognized flavivirus threat

PONE-D-25-54301R1

Dear Dr. Myint,

We’re pleased to inform you that your manuscript has been judged scientifically suitable for publication and will be formally accepted for publication once it meets all outstanding technical requirements.

Kind regards,

Mirko Cortese

Academic Editor

PLOS One

---

## [Editor Report · Acceptance letter]

PONE-D-25-54301R1

PLOS One

Dear Dr. Myint,

I'm pleased to inform you that your manuscript has been deemed suitable for publication in PLOS One. Congratulations! Your manuscript is now being handed over to our production team.

Kind regards,

on behalf of

Prof. Mirko Cortese

Academic Editor

PLOS One